# Immunomodulation in Pancreatic Cancer

**DOI:** 10.3390/cancers12113340

**Published:** 2020-11-12

**Authors:** Mithunah Krishnamoorthy, John G. Lenehan, Jeremy P. Burton, Saman Maleki Vareki

**Affiliations:** 1Department of Microbiology and Immunology, University of Western Ontario, London, ON N6A 3K7, Canada; mithunah.krishnamoorthy@lhsc.on.ca (M.K.); Jeremy.Burton@lawsonresearch.com (J.P.B.); 2Cancer Research Laboratory Program, Lawson Health Research Institute, London, ON N6A 5W9, Canada; 3Division of Medical Oncology, Department of Oncology, University of Western Ontario, London, ON N6A 3K7, Canada; John.lenehan@lhsc.on.ca; 4Canadian Centre for Human Microbiome and Probiotics, Lawson Health Research Institute, London, ONN6A 4V2, Canada; 5Division of Urology, Department of Surgery, University of Western Ontario, London, ON N6A 3K7, Canada; 6Department of Pathology and Laboratory Medicine, University of Western Ontario, London, ON N6A 3K7, Canada; 7Division of Experimental Oncology, Department of Oncology, University of Western Ontario, London, ON N6A 3K7, Canada

**Keywords:** pancreatic cancer, pancreatic ductal carcinoma, immunotherapy, anti-PD1, microbiome, fecal microbiota transplant, T-cells

## Abstract

**Simple Summary:**

Pancreatic cancer is an aggressive disease that has few options for treatment. Immune checkpoint blockers have been revolutionary in specific cancer types yet have not shown promise in pancreatic cancers. This review discusses components of the immune system needed for successful anti-tumor responses against pancreatic cancer and how these mechanisms can be exploited to develop new therapeutic strategies.

**Abstract:**

Pancreatic cancer has a high mortality rate, and its incidence is increasing worldwide. The almost universal poor prognosis of pancreatic cancer is partly due to symptoms presenting only at late stages and limited effective treatments. Recently, immune checkpoint blockade inhibitors have drastically improved patient survival in metastatic and advanced settings in certain cancers. Unfortunately, these therapies are ineffective in pancreatic cancer. However, tumor biopsies from long-term survivors of pancreatic cancer are more likely to be infiltrated by cytotoxic T-cells and certain species of bacteria that activate T-cells. These observations suggest that T-cell activation is essential for anti-tumor immunity in pancreatic cancers. This review discusses the immunological mechanisms responsible for effective anti-tumor immunity and how immune-based strategies can be exploited to develop new pancreatic cancer treatments.

## 1. Introduction

The majority of pancreatic cancer patients have a poor prognosis, where the five-year survival rate is 9% in the United States, with an increasing incidence rate of 1.03% per year [1,2]. Pancreatic ductal adenocarcinoma (PDAC) is an aggressive form of pancreatic cancer that makes up 90% of all diagnosed cases of pancreatic cancer [3]. Other types of pancreatic cancer, such as neuroendocrine tumors, which secrete insulin, or acinar carcinomas, which release digestive enzymes, are less common. PDAC develops from neoplasms of the cells lining the pancreatic ducts and usually presents without symptoms until advanced stages of the disease [4].

At this time, there are limited treatment options for patients with advanced and incurable PDAC, all of which are chemotherapy based. The FOLFIRINOX regimen (a combination of oxaliplatin, irinotecan, leucovorin, and fluorouracil) is often used as the first line of treatment for patients who are likely to be able to tolerate the relatively high toxicity [5,6]. The median overall survival for this regimen is 11.1 months [6]. Patients with comorbidities are often treated with gemcitabine alone or in combination with albumin-bound (nab)-paclitaxel with a median overall survival of only approximately 6.8 and 8.5 months, respectively [5,6,7]. Surgical resection remains the only potentially curative option, with a median overall survival of 54.4 months when combined with adjuvant FOLFIRINOX [8]. However, only about 20% of patients are considered resectable at the time of diagnosis [9].

Immunotherapy with immune checkpoint blockade (ICB) has been a breakthrough in treating some of the most treatment-refractory cancers, including advanced melanoma and metastatic non-small-cell lung carcinoma [10,11]. However, ICBs did not show efficacy in PDAC, and tumors showed primary resistance to monotherapy with ipilimumab (an anticytotoxic T lymphocyte antigen 4) or atezolizumab (anti-programmed cell death receptor ligand 1) [10,12]. Currently, the only approved immunotherapy for pancreatic cancer is pembrolizumab (anti-PD1), recommended for use in patients with mismatch repair deficiencies (dMMR) or microsatellite instability (MSI)-high tumors as a second line of treatment [13,14]. MSI-high tumors often have a higher tumor mutation burden and a more T-cell-inflamed microenvironment that is not typically seen in the immune-hostile PDAC tumor microenvironment. Modifying the tumor microenvironment to create a more T-cell-inflamed microenvironment might improve PDAC response to immunotherapy, either alone or in combination with other therapies. 

## 2. The Tumor Microenvironment

PDAC tumors are unique from other solid tumors as they are surrounded by an extremely dense network of proliferating fibroblastic cells and extracellular matrix (ECM) known as desmoplasia. In a trial of 23 PDAC patients undergoing resection, an increase of desmoplasia in biopsies was correlated with shorter overall survival [15]. This dense network of cells can compress blood vessels reducing drug perfusion into the tumor and develop a hypoxic microenvironment [16,17]. Interestingly, the ECM is a source of nutrients for PDAC cells [18]. However, the depletion of stromal cells that excrete ECM only results in more aggressive disease [19,20]. For example, depletion of myofibroblasts in mice resulted in the accumulation of regulatory T-cells (Tregs) and reduced overall survival [20]. This suggests that the stromal cells also have some anti-tumor functions.

One of the cell types that predominantly form the ECM is pancreatic stellate cells (PSC), also referred to as cancer-associated fibroblasts [4]. PSCs are star-shaped cells that, when activated, can facilitate the formation of desmoplasia. They also contribute to inhibiting cancer apoptosis and can potentially travel to distant sites from the primary tumor to initiate the growth of metastasis [4,21]. In addition to directly supporting tumor cell growth, PSCs also play a role in immunosuppression. PSCs can secrete proinflammatory cytokines such as IL-6, promoting the expansion and function of myeloid-derived suppressor cells (MDSCs) [22]. PSCs were also found to sequester CD8^+^ T-cells and prevent them from entering into the tumor [23]. As CD8^+^ T-cells are essential for anti-tumor immunity, blocking their entrance into PDAC tumors can contribute to the formation of an immunologically cold microenvironment, reducing immunotherapy response. A meta-analysis of PDAC patients has shown that those with higher CD8^+^ T-cell infiltration have better overall survival [24]. 

PSCs capable of sequestering T-cells were found to attract CD8^+^ T-cells by expressing the chemoattractant CXCL12. Interestingly, PDAC patients have higher expression of the receptor for CXCL12/CXCR4 on their T-cells compared to healthy volunteers [23]. Blocking the CXCL12/CXCR4 interaction reduced tumor burden in preclinical models when used in combination with anti-PDL1 [25]. However, clinical trials combining a CXCR4 antagonist with pembrolizumab failed to reduce patients’ tumor burden during treatment [26]. Since it is not currently clear whether the abundance of PSCs impacts clinical outcomes, further understanding of these cells’ roles is required [4].

Type 1 conventional dendritic cells (cDC1) are central in activating naïve T-cells. These antigen-presenting cells also decreased in abundance as the pancreatic tissue progressed from healthy tissue to premalignant and then malignant [27]. Additionally, the remaining cDC1 failed to mature. As cDC1s are crucial for presenting antigenic peptides to prime T-cells for anti-tumor immunity, the lack of cDC1 in PDAC severely reduces T-cells’ ability to mount a response. One study showed that increased PDAC progression was correlated with reduced frequency and functionality of cDC1s. However, the administration of Fms-related tyrosine kinase 3 ligand (Flt3L) early in disease progression could increase cDC1 infiltration into the tumor and increase anti-tumor immunity. Flt3L regulates cDC development from myeloid progenitors and, when combined with CD40, can stabilize tumor growth in a murine model [28,29]. Notably, the administration of the anti-CD40 antibody as a single agent was found to increase the frequency of cDC1 and increase maturation markers in tumor-draining lymph nodes in preclinical models [27]. CD40 is a cell surface molecule that regulates antigen-presenting cells’ activation by binding to its natural ligand, CD40, on T-cells. Anti-CD40L can agonistically bind, circumventing the need for T-cells to activate antigen-presenting cells [30]. 

The unique stroma of PDAC allows for the differential recruitment of lymphocytes. While cells needed for anti-tumor immunity are present in lower abundance, immunosuppressive cells are found in higher numbers. Tregs play a significant role in suppressing the anti-tumor response in PDAC. These FOXP3^+^CD25^+^ Tregs bind to DCs and prevent them from activating CD8^+^ T-cells. They also secrete immunosuppressive cytokines such as IL-10, which suppresses DC activation. Higher levels of Treg infiltration into PDAC tumors are associated with worse overall survival in patients [24]. Whether Tregs’ depletion would be an effective treatment strategy to establish a robust anti-tumor immunity in PDAC is still being tested in preclinical models. One study showed that depleting Tregs in mice improved vaccine efficiency and recruited more tumor antigen-specific CD8^+^ T-cells into pancreatic tumors [31]. In another study, Tregs depletion improved the efficacy of a listeria-based vaccine in preventing the progression of precancerous lesions into PDAC in a mouse model. However, this experimental combination therapy did not prevent advanced PDAC from further progression [32]. At least one study found that the depletion of Tregs accelerated pancreatic carcinogenesis in mouse models. This study showed that Tregs’ depletion led to an increase of MDSCs in the tumor microenvironment, establishing an even more immunosuppressive microenvironment [33]. 

PDAC tumors are also well infiltrated by MDSCs, which is a hallmark of pancreatic cancer. MDSCs are a heterogeneous group of immature myeloid cells that inhibit T-cell proliferation and migration within the tumor microenvironment by releasing reactive oxygen species and nitrogen species. MDSCs can induce T-cell tolerance by expressing inhibitory ligands such as PD-L1 and CTLA-4. They also increase Tregs abundance by engaging CD40 in the presence of IL-10 and transforming growth factor-beta (TGF-β) [34]. To test the effect of myeloid cell depletion on tumor progression, a mouse model of PDAC was crossed with CD11b-diptheria toxin receptor-expressing mice to facilitate inducible deletion of macrophages and MDSCs. The depletion of MDSCs prevented the formation of precancerous lesions that may ultimately progress to an invasive tumor, and in already established tumors, MDSC depletion led to arrested tumor growth. In the absence of MDSCs, tumors underwent increased CD8^+^ T-cell mediated apoptosis. MDSCs were found to promote the expression of PD-L1 in tumor cells in an epidermal growth factor receptor (EGFR)/mitogen-activated protein kinase (MAPK)-dependent manner [35]. This suggests that CD8^+^ T-cell activity may be improved by depleting MDSC through MEK inhibitors. A phase I clinical trial attempted to deplete MDSCs by administering an antagonistic TNF-related apoptosis induced ligand-receptor 2 (TRAIL-R2) antibody in patients with various metastatic cancers, including pancreatic cancer. Treated patients had reduced numbers of MDSCs in peripheral blood samples compared to pretreatment numbers; furthermore, 50% of patients had reduced MDSCs in tumor biopsies compared to pretreatment numbers. As this treatment was well-tolerated with only mild to moderate adverse events, a TRAIL-R2 antagonist may be a suitable candidate for combination with other immunotherapies [36].

Tumor-associated macrophages (TAMs) also dominate the PDAC tumor microenvironment and display immunosuppressive functions [37]. The presence of these macrophages in higher numbers was associated with poor overall survival [38]. Notably, TAMs release cytokines and growth factors that promote immune evasion, angiogenesis, and metastasis, promoting aggressive cancer progression [39]. Therefore, reducing the number of TAMs would be beneficial in PDAC treatment. Indeed, targeting TAMs by using an inhibitor of colony-stimulating factor 1 receptor (CSF 1R) reduced the size of existing tumors and increased overall survival in preclinical models. This was accompanied by an increase in T-cell activation and decreased levels of immune suppressive cytokines such as IL-10 [40]. Blocking CSF1R also resulted in increased efficacy of anti-PD1 and anti-CTLA4 checkpoint blockade [41].

## 3. Tumor Microbiome

While the gut microbiome remains the most studied habitat for microorganisms, unique tumor type-specific bacteria have been found in various tumors, including PDAC [42]. Bacteria play a role in the breakdown of xenobiotics, which may, in part, explain PDAC resistance to certain chemotherapies. An in vitro study has shown that the class of gammaproteobacteria can metabolize gemcitabine into an inactive form, which may explain gemcitabine resistance in some PDAC patients. To validate this result, the bacterial composition of human PDAC samples were assessed using 16S rRNA sequencing. In this study, 76% of PDAC tumor samples were positive for bacterial DNA, with the majority being from gammaproteobacterial [43]. 

While PDAC has a low survival rate, some patients live past five years after the initial diagnosis. A study by Riquelme et al. showed that this might be determined partly due to the PDAC tumor microbiome. Long-term survivors of the disease with a median survival of 10.1 years were found to have higher diversity in their tumor microbiome compared to short-term survivors (median survival of 1.5 years). These long-term survivors had a distinct bacterial signature consisting of Pseudoxanthomonas/Streptomyces/Saccharopolyspora/Bacillus clausii. This bacterial signature effectively predicted those who would survive long-term in both discovery and validation cohorts and was associated with better cytotoxic T-cell responses. Long-term survivors had twice as many infiltrating CD8^+^ T-cells expressing granzyme B compared to short-term survivors. T-cell infiltration was also positively correlated with microbiome diversity. Notably, this study showed that the pancreatic tumor microbiome could potentially be influenced by the gut microbiome [44]. While the tumor shared ~25% of the same operational taxonomic units (OTUs) with the gut, most of the tumor microbiome was unique to the pancreas. This suggests that the pancreatic tumor provides a unique environment to colonize certain bacteria.

When human fecal material was transferred into a murine model via oral gavage, 40% of the murine gut microbiome was composed of bacteria of human origin. Only 5% of the tumor microbiome harbored bacteria of human origin [44]. Nevertheless, this shows that certain bacteria from a fecal transplant could eventually translocate from the gut to the pancreatic tumor. The gut microbiome has been found to play an essential role in modulating the response to immunotherapy. Patients who responded to ICBs were found to have a different gut microbiome composition than nonresponding patients. Thus, modifying the gut microbiome in PDAC patients may be a feasible treatment strategy in the future [45,46]. Indeed, we and others are testing the safety of combining modifying gut microbiome via fecal microbiota transplants (FMT) to standard immunotherapy regimens in various solid tumors (NCT03772899, NCT03353402). 

Other studies have deemed the tumor microbiota to have a carcinogenic effect. The use of ablative broad-spectrum antibiotics in a slowly progressing pancreatic cancer mouse model showed protection against disease progression. This was due to an increase in CD4^+^ and CD8^+^ T-cell infiltration, as well as a decrease in MDSCs and TAMs in the tumors of antibiotic-treated mice [47]. Further studies are required to determine whether specific bacteria lead to oncogenesis and whether they can be selectively removed to improve PDAC prognosis. 

## 4. Biomarkers of Long-Term Survivors

Long-term survivors represent a small proportion of PDAC patients. Determining why these patients have a better prognosis may provide valuable insight for designing novel treatments that can mimic some of the features present in tumors of these long-term survivors. Using immunohistochemistry and whole-exome sequencing on PDAC biopsies, a study showed high CD8^+^ T-cell tumor infiltration and highly immunogenic, newly formed T-cell epitopes, neoantigens, found in tumors correlated with better survival in PDAC patients. Neither factor on its own was associated with survival. This highlights the importance of the specific neoantigens present within the tumor, rather than the quantity of each.

Interestingly, high-quality neoantigens in long-term survivors were found to be homologous with pathogenic microbial peptides and may be used as a biomarker for better survival [48]. As mentioned previously, a specific bacterial signature in the tumor microbiome was also indicative of long-term PDAC survival [44]. This suggests that certain bacteria may be used as a therapeutic vaccine to induce T-cell activation in immunotherapy-refractory cancers, as many of these long-term survivors harbor T-cells that cross-react to microbial peptides and tumor-derived neoantigens [48]. 

Many other cell types promote an inflammatory response. For example, group 2 innate lymphoid cells (ILC2), which are innate antigen-independent lymphocytes, infiltrated PDAC tumors. Higher frequencies of ILC2s were correlated to longer survival periods [49]. Interestingly, IL-33 was found to be important in the activation of ILC2s, and patients with higher levels of IL-33 mRNA transcripts were found to live longer than 10 years. Like exhausted T-cells, ILC2 also express PD1, which attenuates their effector functions. It was PD1 blockade therapy that increases ILC2 activity and thus increases the anti-tumor response [49].

## 5. Adjuvant and Neoadjuvant Therapies

Surgical resection is the best curative option for PDAC patients, although ultimately, 80% of those who undergo surgery recur, either due to micrometastases or microscopic residual disease in the tumor bed [1,50]. The current standard of care for upfront resectable PDAC is surgery followed by adjuvant combination chemotherapy [51]. The administration of gemcitabine and capecitabine after surgery resulted in median overall survival of 28.0 months, a modest increase compared to gemcitabine monotherapy with a median overall survival of 25.5 months [52]. The majority of patients in this trial had lymph node involvement. The efficacy of immunotherapy has also been tested in both neoadjuvant and adjuvant settings in other tumor types. Neoadjuvant immunotherapy has shown great promise primarily by reinvigorating exhausted T-cells before surgery. Tumor-specific CD8^+^ T-cells can be primed by antigens derived from the primary tumor and can continue to exert their cytotoxic effects on micrometastases once the primary tumor is removed [53]. While univariate analysis showed that neoadjuvant immunotherapy resulted in better overall survival than adjuvant immunotherapy in pancreatic cancer patients, multivariate analysis showed that those treatment strategies were not significantly different [54].

Interestingly, adjuvant immunotherapy combined with chemotherapy was more effective than chemotherapy alone [55]. The combination may be effective, as chemotherapy can ablate tumors and release tumor antigens in the process. These antigens can then prime T-cells. Furthermore, treatment strategies reducing desmoplasia allows T-cells to infiltrate the tumor microenvironment more effectively [23]. Administration of nab-paclitaxel was found to reduce the density of the desmoplastic stroma in preclinical models and facilitated the delivery of gemcitabine to the tumor [56]. 

The combination of radiation and chemotherapy was also effective in stimulating an anti-tumor immune response. Neoadjuvant chemoradiation given as 30 Gy of radiation combined with gemcitabine and a fluoropyrimidine derivative was found to increase the infiltration of both CD4^+^ and CD8^+^ T-cells. As expected, increased CD8^+^ T-cell infiltration of the tumors was associated with prolonged overall survival and lower recurrence rates in patients with resectable pancreatic cancer [57]. Another study assessing the effect of neoadjuvant chemoradiation showed no difference in the recruitment of CD4^+^ or CD8^+^ T-cells; however, a decrease in FOXP3^+^ Tregs was observed [58]. More studies are required to determine if T-cells’ recruitment can be further improved by optimizing the timing of treatment and surgery. 

## 6. Combination Strategies

PDAC is a complex disease with several cellular components that can either hinder or promote anti-tumor immunity. Therefore, successful therapeutic strategies may combine several existing therapies. For example, a combination of different ICBs to chemotherapy has shown success in lung cancer. Chemotherapy can induce apoptosis of tumor cells, and in the process, release tumor antigens that can activate anti-tumor immunity. A randomized phase II trial (NCT0287931) compared the use of gemcitabine and nab-paclitaxel against gemcitabine and nab-paclitaxel combined with durvalumab (anti-PDL1) and tremelimumab (anti-CTLA-4). Of the 11 patients that received gemcitabine, nab-paclitaxel, durvalumab, and tremelimumab, 8 achieved a partial response, showing the potential of combination therapy [59]. A CD40 agonist is being combined with gemcitabine, nab-paclitaxel, and nivolumab (anti-PD1) in a phase 1b clinical trial (NCT02482168) in patients with metastatic PDAC. Of 24 patients, 14 had a partial response [60]. Currently, this trial has progressed to a phase II study (NCT03214250).

## 7. T-Cell Activation Strategies for PDAC Treatment

With the modest progress made with chemotherapy, radiation, and ICBs, novel treatment strategies for pancreatic cancer are actively being explored. Considering that CD8^+^ T-cells play a large role in the long-term survival of PDAC patients, approaches often involve activation of the anti-tumor immune response.

### 7.1. Chimeric Antigen Receptor (CAR) T-Cells

CAR T-cell therapy is an adoptive cell therapy, where autologous T-cells are isolated and genetically engineered to express receptors that bind to tumor antigens. These modified T-cells are then reinfused into the patient. When CAR T-cells recognize target antigens, they become activated and exert cytotoxic effects on the tumor cell (Figure 1a). Currently, CAR T-cell therapy has only been approved by the FDA for hematologic malignancies [61]. The safety of a mesothelin-specific CAR T-cell therapy was tested in six patients with chemotherapy-refractory metastatic PDAC. These patients did not experience a cytokine storm or dose-limiting toxicities. However, only two patients achieved stable disease with overall survival of 3.8 and 5.4 months [62]. Preclinical studies are also testing the efficacy of switchable CAR T-cells against human epithelial growth factor receptor 2 (HER2) antigen. Switchable CAR T-cells are composed of two parts. For the first part, the autologous T-cells are engineered with the signaling portion of the T-cell receptor and the second part consists of a “switch” protein that binds to both the signaling portion of the autologous T-cells and the tumor antigen. This allows the fine-tuning of the T-cell response by administering variable concentrations of the switch protein. The switch CAR T-cell against HER2 antigen led to the complete remission of xenograft models derived from patients with stage IV PDAC [63]. 

Unfortunately, engineered T-cells become progressively dysfunctional within the tumor and require repeated reinfusions to achieve a therapeutic benefit in experimental models [64]. One study attempted to mitigate this process by modulating TAM function. The administration of agonistic anti-CD40 along with CAR T-cells resulted in increased proliferation and cytotoxic abilities of CAR T-cells [65]. Notably, while the depletion of TAMs resulted in increased proliferation of endogenous T-cells, there was no marked increase in CAR T-cell infiltration of the tumor [65]. The efficacy of CAR T-cell function can also be improved by coexpressing an engineered immunomodulatory fusion protein. A fusion protein consisting of the inhibitory Fas receptor ectodomain and a stimulatory 4-1BB endodomain could convert inhibitory signals into activating ones. The use of CAR T-cells expressing the Fas-4-1BB fusion protein was found to significantly improve survival in an aggressive mouse model of pancreatic cancer [66]. This suggests that several methods can be developed to improve the efficacy of current CAR T-cell technology to treat pancreatic cancers. 

### 7.2. Microbiome Modification

As mentioned previously, the tumor microbiome can predict the overall survival of PDAC patients. This suggests that modification of the microbiome can potentially increase the response to therapeutics. The microbiome can be altered either by antibiotics, FMT, or defined bacterial consortia (Figure 1b). 

#### 7.2.1. Antibiotics

Recently, the use of antibiotics has been shown to abrogate the effects of anti-PD1 immunotherapy in both melanoma and renal cell carcinoma [45]. However, the opposite was found to be true by Pushalkar et al. [47] in regard to PDAC. An increase in bacteria was found in the cancerous pancreas of mice compared to healthy pancreas. Decreasing bacterial abundance using broad-spectrum antibiotics was successful in slowing tumor growth in KC mice (spontaneous pancreatic neoplasia model). This growth was further attenuated by combining broad-spectrum antibiotics with anti-PD1. The use of antibiotics was found to increase the intratumoral CD8/CD4 T-cell ratio, which may explain the reduction in tumor growth (Figure 2a) [47]. Antibiotic treatment strategies could be employed in neoadjuvant settings to abrogate harmful bacteria before surgery. More studies are required to validate this strategy. 

#### 7.2.2. Fecal Microbiota Transplantation

FMT is commonly used to treat dysbiosis of the gut with a high success rate, such as with refractory Clostridium difficile infections. Presumably, FMT can be used to treat other conditions with microbial dysbiosis, including PDAC [44,67]. Akkermansia muciniphila, a commensal bacterium commonly found in healthy stool, was found to induce increased infiltration of CD4^+^ T-cells in RET melanomas when combined with the anti-PD1 antibody [45]. This suggests that certain bacteria found in healthy stool samples may boost immunity by activating certain populations of T-cells. Modifying the gut microbiome via FMT may also be an attractive strategy in the neoadjuvant setting for PDAC (Figure 2b).

#### 7.2.3. Bacterial Consortia

The gut is colonized by various bacterial populations, some of which have a capacity to induce T-cell activation (Figure 2c). A consortium of 11 bacteria was found to increase the frequency of INFγ^+^-producing CD8^+^ T-cells in the intestines of germ-free mice. This consortium consisted of seven Bacteroidales and four non-Bacteroidales species, all of which needed to be administered in combination to achieve maximal T-cell activation. When administered with an anti-PD1 or anti-CTLA 4, the mix of 11 bacteria slowed tumor growth in mice [68]. This suggests that a mix of bacteria may be taken as a probiotic to improve the efficacy of immunotherapy. Whether this would benefit PDAC patients still needs to be explored. 

### 7.3. Oncolytic Vaccine Combinations

Oncologic vaccines work to expose the host immune system to tumor-specific antigens. To date, vaccines on their own have shown no clinical benefits in pancreatic cancer [69]. However, there are some promising results indicative of inducing an anti-tumor immune response when vaccines are combined with chemotherapy. An irradiated, granulocyte-macrophage colony-stimulating factor (GM-CSF), secreting, allogeneic PDAC vaccine (GVAX) was given as a single agent or in combination with low-dose cyclophosphamide to patients with resectable PDAC. After two weeks of vaccine administration, tertiary lymphoid organs were found inside tumor biopsies. This indicates that lymphocytes are actively infiltrating the tumor microenvironment and facilitating an adaptive response [70]. 

## 8. Conclusions

PDAC is an aggressive cancer with limited treatment options that provide only modest benefits. Even with the advent of immunotherapy, response rates in PDAC are very low. However, new technologies are being developed to improve patient outcomes and quality of life. Patients who are long-term survivors were found to have an increase in the number of neoantigens and a unique tumor microbiome, both of which correlated with increased T-cell infiltration. This suggests that successful treatment regimens should aim to increase CD8^+^ T-cell infiltration. Finally, the response to immunotherapy as a single agent is low in PDAC patients and its efficacy may be increased by modifying the tumor environment or microbiota. 

## Figures and Tables

**Figure 1 cancers-12-03340-f001:**
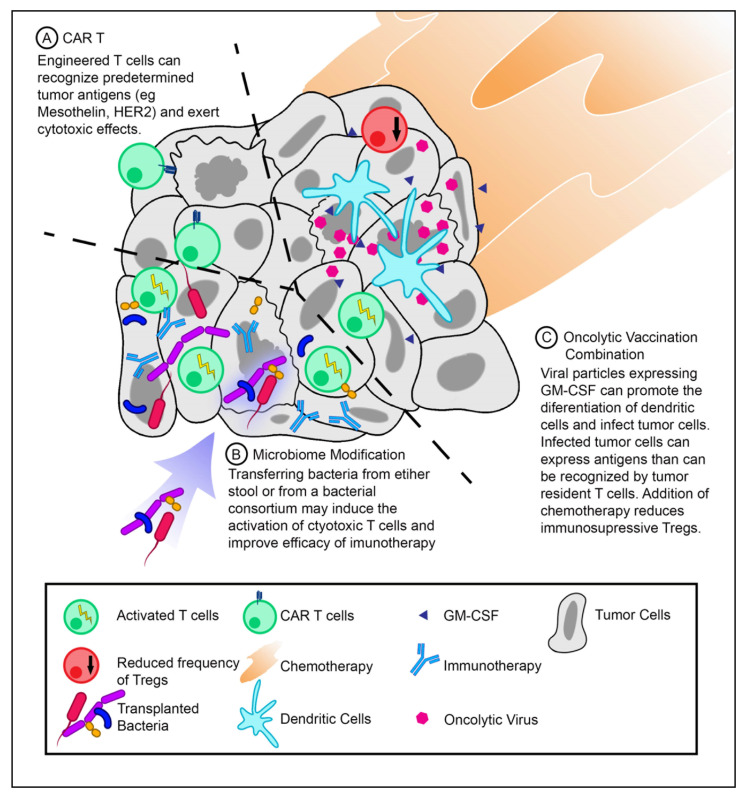
T-cell activation strategies for pancreatic ductal adenocarcinoma (PDAC) treatment. (**A**) Infusion of chimeric antigen receptor T-cells (CAR T-cells) against PDAC tumor antigens. CAR T-cells can recognize antigens specific for their CARs and exert cytotoxic effects. (**B**) Modifying the tumor microbiome may introduce novel antigens that can activate T-cells and improve the efficacy of immunotherapeutics. (**C**) Therapeutic vaccination with oncolytic viruses expressing granulocyte–macrophage colony-stimulating factor (GM-CSF) in combination with chemotherapy (low-dose cyclophosphamide) can increase the frequency of dendritic cells while decreasing the frequency of Tregs.

**Figure 2 cancers-12-03340-f002:**
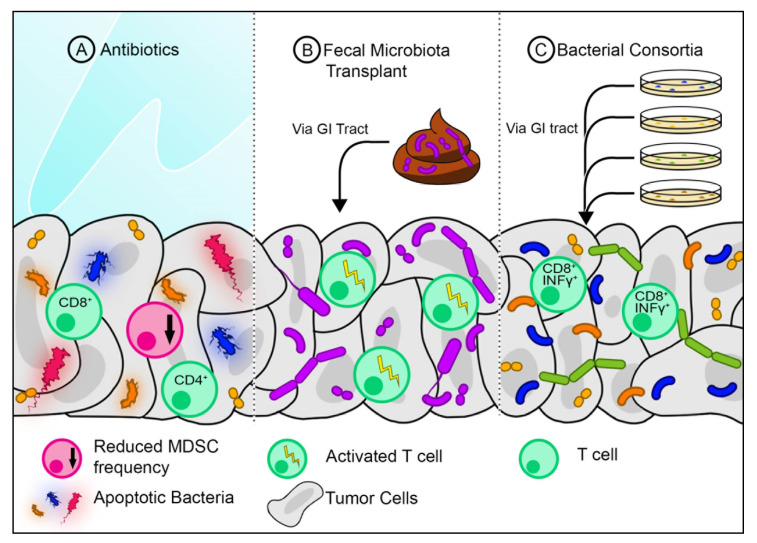
Strategies for modifying the tumor microbiome. (**A**) Antibiotics eliminate certain bacteria and may reduce frequencies of myeloid-derived suppressor cells (MDSC) while also increasing the frequency of CD4^+^ and CD8^+^ T-cells. (**B**) Fecal microbiota transplants from healthy individuals can introduce new bacteria into the gut, where in addition to re-establishing gut microbiome diversity, some of the newly introduced bacteria may translocate to the tumor. These bacteria may activate certain T-cell populations. (**C**) A Consortium of bacteria of known species grown in vitro can be introduced to the gut where it can translocate to the tumor. Certain combinations of bacterial species can activate anti-tumor CD8^+^ T-cells.

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
