# Peer review of "Immunomodulation in Pancreatic Cancer"

_cancers, 2020, doi:10.3390/cancers12113340_

Round 1

Reviewer 1 Report

The review by Krishnamoorthy et al. entitled, " IMMUNOMODULATION IN PANCREATIC CANCER" summarizes the key points of the immunosuppressive tumor microenvironment of pancreatic cancer and discussed possible immune regulatory approaches to activate anti-tumor immune responses. This is a timely and important topic for review as pancreatic cancer is one of the most lethal malignant tumor types with limited treatment options. Also, it is one cancer type  that has not benefited from immunotherapies, making this another example of the lack of treatment options for pancreatic cancer patients. The authors' focused mainly on the most recent literature describing the mechanisms behind this suppression of the anti-tumor immune response and strategies for increasing pancreatic tumor immunogenicity. Thus, this review will have import messages to both basic and clinical immunologists. However, the  manuscript could be greatly improved with further information on tumor associated macrophages and fibroblasts, which are two main stromal cell populations within pancreatic cancer that play significant immune regulatory roles. At present, many strategies targeting tumor associated macrophages or fibroblasts are under development or being tested in the clinics, combining chemotherapy or checkpoint blockade. Adding these additional topics to the review would give greater impact and certainly make it be of interest to broader audience.

Author Response

Reviewer #1

The review by Krishnamoorthy et al. entitled, " IMMUNOMODULATION IN PANCREATIC CANCER" summarizes the key points of the immunosuppressive tumor microenvironment of pancreatic cancer and discussed possible immune regulatory approaches to activate anti-tumor immune responses. This is a timely and important topic for review as pancreatic cancer is one of the most lethal malignant tumor types with limited treatment options. Also, it is one cancer type that has not benefited from immunotherapies, making this another example of the lack of treatment options for pancreatic cancer patients. The authors' focused mainly on the most recent literature describing the mechanisms behind this suppression of the anti-tumor immune response and strategies for increasing pancreatic tumor immunogenicity. Thus, this review will have import messages to both basic and clinical immunologists.

We thank the reviewer for their positive comments

However, the manuscript could be greatly improved with further information on tumor associated macrophages and fibroblasts, which are two main stromal cell populations within pancreatic cancer that play significant immune regulatory roles. At present, many strategies targeting tumor associated macrophages or fibroblasts are under development or being tested in the clinics, combining chemotherapy or checkpoint blockade. Adding these additional topics to the review would give greater impact and certainly make it be of interest to broader audience.

We thank the reviewer for this suggestion. We have added a section on cancer associated fibroblasts (Pancreatic Stellate cells) and the role they play in the tumor microenvironment. (see line 125-136). We have also included studies that explore the role on Tumor associated macrophages in PDAC (see lines 443-452, marked up version).

Reviewer 2 Report

Excellent review of an important topic.  Minor suggestion:

1.  CAR T-cell therapy section.  Line 234.  It is noted that 2 patient achieved stable disease.  It is important for authors to note how many patients were included in this study, so that we know 2 of how many patients had stable disease.

2.  Pancreatic Stellate Cells (PSC) - found it strange that the authors have not included any section focused on PSCs and their role in immunomodulation.  I think this is important.

Author Response

Reviewer #2

  1. CAR T-cell therapy section.  Line 234.  It is noted that 2 patient achieved stable disease.  It is important for authors to note how many patients were included in this study, so that we know 2 of how many patients had stable disease.

We thank the reviewer for this suggestion. This study had a total of 6 patients and is now indicated on line 632 (marked up version).

  1. Pancreatic Stellate Cells (PSC) - found it strange that the authors have not included any section focused on PSCs and their role in immunomodulation.  I think this is important.

We previously included information regarding PSCs and their role in sequestering CD8+ T cells and how this phenomenon could be mitigated using CXCL12/CXCR4 blockade (See previous non marked up version line 141, marked up version). We now have included more information regarding PSCs (See line 180-187, marked up version).

Reviewer 3 Report

Krishnamoorthy et al. provided a concise overview of the current literature concerning antitumor immunity and clinical results of immunotherapeutic studies in pancreatic ductal adenocarcinoma (PDAC). Acknowledging the tumor microbiome is highly appreciated as more and more evidence substantiates the importance of gut and intra-tumoral bacterial composition in order to initiate effective immunological responses (Helmink et al, Nat Med; 2019, Riquelme et al, Cell, 2019). However, the introduction of several additional studies may support certain topics and improve the current manuscript.

  1. Rule 67-77

Also add studies that focus on the depletion of extracellular matrix. Previously targeting fibroblast activation protein (FAP) in PDAC did not show meaningful results (Gunderson eta l, PLoS One, 2019; Nugent eta al, J. Clin. Oncol. 2007). Also, depletion of aSMA+ fibroblasts led to disease progression (Ozdemir et al, Cancer Cell 2014). This demonstrates that the tumor biology in the pancreas is highly complex, and that stroma may also have cancer-restraining functions. This may also explain the failure of clinical studies utilizing PEGPH20 in order to ablate tumor stroma (Ramanathan et al, J. Clin. Oncol, 2019).

  1. Rule 84-93

Stress the importance of dendritic cells (DCs) in PDAC by adding the findings of DeNardo’s group (Hegde et al, Cancer Cell, 2020). They showed that DC deficiency is correlated with disease progression and restoration of DC function led to improved anti-tumor immunity.

  1. Chapter: The Tumor Microenvironment

The TME of PDAC is intertwined with multiple immunosuppressive cells. Next to Tregs and MDSCs, tumor associated macrophages (TAMs) are abundantly present and able to exert immunosuppressive functions in PDAC. Tumor involution is observed after reprogramming of TAMs (Candido et al, Cell Rep, 2018; Zhu et al, Cancer Res, 2014). Please also briefly discuss the role of TAMs in PDAC in this chapter.

  1. As mentioned in rule 71, the presence of thick desmoplastic stroma may exclude effector T cells. Therefore, a multimodal treatment regime may be a prerequisite for a robust anti-tumor response in PDAC. Add studies demonstrating rational combination strategies and promising results for PDAC. Vonderheide’s group demonstrated encouraging results when metastasized PDAC patients were treated with a regime consisting of chemotherapy, CD40 agonist and immune checkpoint blockers (NCT03214250, PRINCE Trial). In addition, rationally combining DC therapy and a CD40 agonist demonstrated promising results in a preclinical model (Lau et al, J Immunother Cancer, 2020).

5. Add journal to reference 22 ( J. Exp. Med.), 52 (Cancer. Immunol. Res.), 53 (J. Exp. Me

Author Response

Reviewer #3

Krishnamoorthy et al. provided a concise overview of the current literature concerning antitumor immunity and clinical results of immunotherapeutic studies in pancreatic ductal adenocarcinoma (PDAC). Acknowledging the tumor microbiome is highly appreciated as more and more evidence substantiates the importance of gut and intra-tumoral bacterial composition in order to initiate effective immunological responses (Helmink et al, Nat Med; 2019, Riquelme et al, Cell, 2019). However, the introduction of several additional studies may support certain topics and improve the current manuscript.

We thank the reviewer for their positive comments.

  1. Rule 67-77

Also add studies that focus on the depletion of extracellular matrix. Previously targeting fibroblast activation protein (FAP) in PDAC did not show meaningful results (Gunderson eta l, PLoS One, 2019; Nugent eta al, J. Clin. Oncol. 2007). Also, depletion of aSMA+ fibroblasts led to disease progression (Ozdemir et al, Cancer Cell 2014). This demonstrates that the tumor biology in the pancreas is highly complex, and that stroma may also have cancer-restraining functions. This may also explain the failure of clinical studies utilizing PEGPH20 in order to ablate tumor stroma (Ramanathan et al, J. Clin. Oncol, 2019).

We have included studies that explore the effect of depleting the extracellular matrix as suggested (see line 176-180, marked up version).

  1. Rule 84-93

Stress the importance of dendritic cells (DCs) in PDAC by adding the findings of DeNardo’s group (Hegde et al, Cancer Cell, 2020). They showed that DC deficiency is correlated with disease progression and restoration of DC function led to improved anti-tumor immunity.

We thank the reviewer for this suggestion. We have expanded on the importance of DCs by including the suggested study (see lines 261-266).

  1. Chapter: The Tumor Microenvironment

The TME of PDAC is intertwined with multiple immunosuppressive cells. Next to Tregs and MDSCs, tumor associated macrophages (TAMs) are abundantly present and able to exert immunosuppressive functions in PDAC. Tumor involution is observed after reprogramming of TAMs (Candido et al, Cell Rep, 2018; Zhu et al, Cancer Res, 2014). Please also briefly discuss the role of TAMs in PDAC in this chapter.

We have included additional information regarding TAMs (see lines 443–452, marked up version).

  1. As mentioned in rule 71, the presence of thick desmoplastic stroma may exclude effector T cells. Therefore, a multimodal treatment regime may be a prerequisite for a robust anti-tumor response in PDAC. Add studies demonstrating rational combination strategies and promising results for PDAC. Vonderheide’s group demonstrated encouraging results when metastasized PDAC patients were treated with a regime consisting of chemotherapy, CD40 agonist and immune checkpoint blockers (NCT03214250, PRINCE Trial). In addition, rationally combining DC therapy and a CD40 agonist demonstrated promising results in a preclinical model (Lau et al, J Immunother Cancer, 2020).

We have discussed 2 clinical trials which show promising results using a combination of chemotherapy and checkpoint inhibitors (see line 608–620, marked up version)

Add journal to reference 22 (J. Exp. Med.), 52 (Cancer. Immunol. Res.), 53 (J. Exp. Me

We thank the reviewer for pointing out this oversight. Journals have been added to the appropriate references
